# Peptide-Based Biomaterials for Bone and Cartilage Regeneration

**DOI:** 10.3390/biomedicines12020313

**Published:** 2024-01-29

**Authors:** Kausik Kapat, Sakshi Kumbhakarn, Rahul Sable, Prashil Gondane, Shruti Takle, Pritiprasanna Maity

**Affiliations:** 1Department of Medical Devices, National Institute of Pharmaceutical Education and Research Kolkata, 168, Maniktala Main Road, Kankurgachi, Kolkata 700054, West Bengal, India; 2Department of Regenerative Medicine and Cell Biology, Medical University of South Carolina, Charleston, SC 29425, USA

**Keywords:** osteochondral, osteogenic, chondrogenic, cartilage, peptide-based, biomaterial

## Abstract

The healing of osteochondral defects (OCDs) that result from injury, osteochondritis, or osteoarthritis and bear lesions in the cartilage and bone, pain, and loss of joint function in middle- and old-age individuals presents challenges to clinical practitioners because of non-regenerative cartilage and the limitations of current therapies. Bioactive peptide-based osteochondral (OC) tissue regeneration is becoming more popular because it does not have the immunogenicity, misfolding, or denaturation problems associated with original proteins. Periodically, reviews are published on the regeneration of bone and cartilage separately; however, none of them addressed the simultaneous healing of these tissues in the complicated heterogeneous environment of the osteochondral (OC) interface. As regulators of cell adhesion, proliferation, differentiation, angiogenesis, immunomodulation, and antibacterial activity, potential therapeutic strategies for OCDs utilizing bone and cartilage-specific peptides should be examined and investigated. The main goal of this review was to study how they contribute to the healing of OCDs, either alone or in conjunction with other peptides and biomaterials.

## 1. Introduction

An articulating joint’s osteochondral (OC) unit comprises the vascularized and mineralized subchondral bone and the acellular and avascular hyaline cartilage connected by a seamless interface [1]. The structural heterogenicity of the OC tissue arises from its diverse organic (cells, aggrecan, collagen) and inorganic (hydroxyapatite) components, along with their spatial orientation, forming gradients from the superficial cartilage to the subchondral bone via middle, deeper, and calcified layers (Figure 1). Cartilage exerts cushioning effects on joint bones and prevents damage from the physiological load; however, an injured cartilage cannot spontaneously regenerate. OCDs are created via a variety of biological (aging, osteochondritis, osteoarthritis) or mechanical (accidental trauma, sports injury, prolonged wear) factors and are characterized by cartilage and bone lesions, severe joint pain, and loss of joint function. They are more common in the middle-aged and older population. The World Health Organization estimates that 595 million people worldwide, or 7.6 percent of the world’s population, suffered from osteoarthritis alone in 2020. This number has increased by 132.2 percent since 1990 [2].

Unfortunately, none of the existing procedures can fully repair an OCD or form hyaline cartilage instead of fibrocartilage [3,4,5,6,7]. OC tissue engineering (TE) seems promising for regenerating functional OC tissue [8]. Growth factors can promote MSCs, osteoblast or chondrocyte recruitment, proliferation and differentiation, vascularization, and maintenance of cartilage homeostasis. However, regulating their target selectivity, dose selection, release kinetics, and spatial distribution is quite challenging. Bioactive peptides are short-chain amino acid sequences that mimic the signaling or binding domains of larger proteins and create a biomimetic environment for recruiting host cells while controlling their activity and differentiation [9]. The ECM or growth-factor-derived peptides that self-assemble into hydrogels were investigated for preparing peptide-enhanced bone graft substitutes or scaffolds for OC regeneration [10,11,12,13]. They are more effective, stable, scalable, and affordable than large proteins, and they do not experience issues like immunogenicity, protein folding, and denaturation [10].

This review’s primary objective was to find promising peptides or peptide-conjugated biomaterials that will help to design a novel multi-peptide-based therapeutic strategy for rapid OCD healing, which has never been done before. After providing a brief overview of the OC unit and the requirements for OC regeneration, this study examined a wide range of bioactive peptides and peptide-conjugated biomaterials that have been reported to have osteo-, angio-, or chondrogenic potential for in vitro and in vivo bone and cartilage regeneration (Table 1). It also covers peptides that help OC regeneration by promoting cell migration and adhesion, assembling into a three-dimensional extracellular environment, or preventing an infection and immunological response. Finally, the review concludes with a summary and outlook.

## 2. Requirements for Osteochondral Tissue Regeneration

The prerequisites of OC regeneration are native tissue mimetic scaffolds, reproducing the heterogeneous structure of the OC tissue, and a well-controlled inflammatory state to counter MMPs-mediated irreversible destruction of the cartilage matrix. The platelet-rich plasma (PRP) and exosome-based therapies, stem cell therapy [98], recombinant IL-1 receptor antagonists, and monoclonal antibodies against tumor necrosis factor-α (TNF-α)-based therapy and genetic engineering have produced variable results. 

On the other hand, tissue engineering, injectable hydrogels, and monolayer scaffolds fail to replicate the intricate spatial distribution of biochemical cues, particularly at the bone and cartilage interface [99,100]. Advanced techniques, such as 3D printing or 3D bioprinting, enable the fabrication of customized scaffolds with structural anisotropy and heterogeneous cell distribution; however, they involve a high cost and intricate setup [101]. Despite their osteogenic and chondrogenic activity, BMPs have less control over chondrocyte hypertrophy and mineralization [98,102]. The localized delivery of bioactive peptides to induce target protein expression; attachment and motility of particular cells; site-specific differentiation; and resist bacterial colonization at the defect site, inflammation, or immunogenicity could be an ideal choice for biomimetic bone and cartilage regeneration. 

## 3. Peptides for Bone Regeneration

Natural bone healing occurs in two different pathways: intramembranous ossification and endochondral ossification [103]. During this process, a variety of cytokines are released [104]. Among them, bone morphogenetic proteins (BMPs, especially BMP-2, -4, and -7) and transforming growth factor-β (TGF-β, especially the β2 and β3 subtypes) play crucial roles in stem cell differentiation by causing hypertrophy and mineralization, while vascular endothelial growth factor (VEGF) and angiopoietins promote neoangiogenesis [105,106]. BMP-2, -3, -4, -7, TNF-α, interferon-γ (IFN-γ), and certain hormones regulate the remodeling phase [107]. Since bioactive peptides follow similar cell-signaling pathways with these proteins (Figure 2a) and may be beneficial for both in vitro and in vivo bone formation, they are grouped into three categories based on their role in osteo-induction, biomineralization, and angiogenesis. 

### 3.1. Osteo-Inducers

#### 3.1.1. Collagen-Mimetic/Derived Peptides

Peptides derived from integrin-binding motifs of collagen type I, which is the most significant ECM protein, are GFOGER, P15, KOD, DGEA, and BCSP1, which can promote osteogenic activity and in vivo bone formation.

Collagen-mimetic GFOGER peptide, which is derived from the collagen α1 chain, selectively promotes α2β1 integrin binding required for osteoblastic differentiation [14]. Besides improving cell attachment, GFOGER successfully induces in vitro osteogenic differentiation and in vivo bone healing [14,67,108,109,110]. GFOGER coating on synthetic PCL scaffolds remarkably enhanced bone formation in critically sized segmental defects in rats by stimulating osteoblast adhesion and differentiation (Figure 2b) [108]. P15, which is a 15-mer peptide derived from the collagen type I α1 chain, has a strong affinity for the cell surface α2β1 integrin receptors. By releasing growth factors and cytokines, the peptide dramatically enhanced the osteogenic differentiation of MSCs [15]. The commercially available P15 formulations significantly enhanced the regeneration of alveolar bone and tibial defects in osteoporotic dogs [111,112]. The collagen-mimetic KOD peptide, which is made of three units, namely, ((PKG)_4_-(POG)_4_-(DOG)_4_), forms a hydrogel through self-assembly inducing platelet activation and blood clotting associated with hematoma formation [113]. POG-based poly-amphiphilic hydrogels allowed for faster recovery (within two weeks) of intervertebral disc defects in rabbits due to a significant increase in ECM deposition [16]. DGEA derived from collagen type I adhesive motif serves as a crucial ligand for osteoblast differentiation [114]. DGEA-containing PA hydrogels seeded with hMSCs substantially upregulated osteogenic markers (OCN, RUNX2, and ALP) [17].

Using a bone- and cartilage-stimulating peptide (BCSP™-1 or NGLPGPIGP) present in human collagen type-I, the proliferation of rat bone-marrow-derived osteoblasts and human or bovine chondrocytes was drastically improved with enhanced the bone mineral density (BMD) and bone mineral content (BMC) in male Wistar rats [18]. Three highly osteogenic peptides (GPAGPHGPVG, APDPFRMY, and TPERYY) derived from tilapia scale collagen hydrolysate notably increased the MC3T3-E1 cell proliferation and mineralization activity (ALP synthesis, osteogenic-related gene expression) at concentrations of 50 μg/mL [19].

#### 3.1.2. BMP-Mimetic/Derived Peptides

As members of the TGF-β superfamily, BMPs are primarily produced by endothelial cells (ECs), osteoblasts, and hypertrophic chondrocytes and can recruit MSCs to the site of injury and differentiate into osteoblasts while inducing ectopic bone formation. 

The KIPKASSVPTELSAISTLYL peptide, which is derived from the knuckle epitope of BMP2, increased the ALP activity of osteoprogenitor cells [115]. P24 is a BMP-2 mimetic peptide with a 24-mer peptide bearing the knuckle epitope of the protein that facilitates binding with BMP receptors. P24 successfully induced ectopic bone formation in rodents [22,23,24]. The PEP7 peptide (CKIPKPSSVP-TELSAISMLYL) derived from BMP-2 promoted adhesion, proliferation, and differentiation of MG-63 cells, as well as new bone formation in a supra-alveolar peri-implant defect model in a micropig mandible [25]. The BMP peptide (KIPKASSVPTELSAISTLYL) derived from BMP2 increased the ALP activity, which is an early marker for bone formation, in murine osteoprogenitor cells [20] and other cell types [116,117,118,119,120], as well as the dose-dependent healing of rabbit radial bone defects [21]. The other osteo-inductive or osteogenic peptides derived from BMP-2 (NSVNSKIPKACCVPTELSAI, KIPKASSVPTELSAISTLYL, DWIVA) produced differential effects on in vitro osteogenic differentiation, as well as ectopic or orthotopic bone formation in vivo [11]. The bone-forming peptide (BFP-2) with a VEHDKEFFHPRYHH sequence, which was isolated from the immature BMP-7 precursor, triggered osteogenic differentiation of BMSCs and induced ectopic bone formation after subcutaneous implantation of BFP-2-treated BMSCs in mice (Figure 2c) [27]. Similarly, the effects of various osteo-inductive peptides derived from BMP-4 (RKKNPNCRRH), BMP-7 (TVPKPSSAPTQLNAISTLYF, GQGFSYPYKAVFSTQ, ETLDGQSINPKLAGL), and BMP-9 (KVGKACCVPTKLSPISVLY) were reviewed [26]. The casein kinase 2 (CK2)-related peptide has a great influence on cell proliferation and apoptosis, and it facilitates in vivo bone formation by interacting with BMP receptor type Ia (BMPRIa) [121]. Three BMP-2 mimetic peptides, namely, CK2.1, CK2.2, and CK2.3, triggered the BMP signaling pathways by inhibiting CK2 binding to BMPRIa [28]. C2C12 cells treated with CK2.3 peptide resulted in osteogenesis, while CK2.2 led to both osteogenesis and adipogenesis [121,122].

#### 3.1.3. Hormone-Derived Peptides 

Parathyroid hormone (PTH) is a major regulator of mineral homeostasis. Parathyroid hormone (PTH)-related peptides called Teriparatide, which are 1–34 peptide domains of PTH (PTH1–34), stimulated osteoblast activity and increased bone density at the fracture site, leading to the healing of non-unions [123,124,125,126,127]. On the other hand, endogenous PTH-related protein (PTHrP) analogs, namely, PTHrP1–34, PTHrP1–36, and PTHrP107–111, increased osteoblast activity and local bone formation [29,30,31]. Calcitonin gene-related peptide (CGRP) is a 37-mer neuropeptide with two isoforms: α- and β-CGRP. They were found to stimulate the proliferation and differentiation of osteoprogenitor cells [32,33,34,35], production of osteogenic molecules like insulin-like growth factor (IGF, especially IGF-1 and -2), BMP-2 [128,129], and reparative bone formation [130].

#### 3.1.4. Circulating Peptides 

Osteogenic growth peptide (OGP), which is a 14-mer peptide occurring in mammalian blood, increases bone formation through anabolic effects on bone cells [131,132] and differentiation of osteoprogenitor cells, leading to upregulated osteogenic markers, including mineralization [36,37,38]. Thrombin peptide 508 (TP508) or Chrysalin is a 23-amino acid peptide and receptor binding domain of thrombin, which enhanced the proliferation, differentiation, and chemotaxis of human osteoblasts [39,40] and VEGF-stimulated angiogenesis [133]. TP508 injected into the fracture gap promoted fracture healing and increased blood vessel formation [61,134,135].

#### 3.1.5. Other ECM-Derived Peptides 

Signaling domains on ECM protein chains are capable of interacting with cell membrane receptors. Various peptides (e.g., FN III9-10/12-14) derived from fibronectin (FN) were shown to promote osteoblast activity and mineralization [41], rabbit calvarial defects healing [136], and augmented BMP-2 and platelet-derived growth factor (PDGF) activities for bone regeneration in vivo [137].

Collagen-binding motif (CBM) is a cleavage product of osteopontin (OPN) that can specifically bind to collagen [138] and promote migration, osteogenic differentiation [139], and bone formation in a rabbit calvarial defect model [42]. The SVVYGLR peptide adjacent to the RGD sequence in OPN significantly enhanced the adhesion and proliferation of MSCs, neovascularization, upregulation of osteogenesis, and angiogenesis when delivered through a collagen sponge [43,44,45]. FHRRIKA, which is a cell-binding and heparin-binding domain of bone sialoprotein (BSP) exerts a favorable effect on osteoblast adhesion, spreading, and mineralization [46]. Higher cell proliferation and viability were observed on rat calvarial osteoblasts that seeded scaffolds containing the RGD and FHRRIKA sequences [140].

### 3.2. Biomineralizing Peptides

Non-collagenous proteins (NCPs), such as dentin sialophosphoprotein, dentin matrix protein 1 (DMP1), and dentin phosphoprotein (DPP), play a significant role in biomineralization. The negatively charged domains (carboxylic acid and phosphate groups) in NCPs serve as preferential sites for the nucleation of hydroxyapatites (HAPs) while stabilizing them into the self-assembled collagen fibrils that act as a template for crystal growth. Peptides derived from such proteins significantly enhance bone formation. 

The Asp–Ser–Ser (DSS) repeating motifs present in DPP have a remarkably strong binding affinity toward calcium ions and HAP [141]. 8DSS, which is a DPP peptide with eight repetitive units of DSS, was the most promising for promoting the mineralization and remineralization of acid-etched enamel [47,48]. Like DSS, 3NSS with three repetitive units of asparagine–serine–serine (aspartic acid in DSS is substituted with asparagine) could remineralize the acid-etched enamel [49]. On the other hand, the DSESSEEDR sequence in dentin matrix protein 1 (DMP1) could bind to demineralized dentin and promote remineralization [50]. The other phosphoprotein-derived peptides, such as SN15, SNA15, DpSpSEEKC, DDDEEK, and DDDEEKC, exhibited high affinity toward HAP [142]. 

Amelogenin, which is found at the dentin–enamel interface, interacts with collagen to control the formation of HAP crystals and their structural alignment [143]. In addition to remineralizing enamel caries, amelogenin-inspired peptides, such as shADP5, QP5, P26, and P32, helped to restore demineralized dentin [51,52]. Since leucine-rich amelogenin peptide (LRAP) is more hydrophilic than amelogenin, the demineralized enamel treated with CS-LRAP hydrogel exhibited quicker nucleation and development of HAP crystals than amelogenin-containing chitosan hydrogel (CS-AMEL) [53]. A non-amelogenin protein called tuftelin is present in tooth enamel and has a role in the mineralization of dental enamel. Tuftelin-derived peptide (TDP) encouraged the remineralization of early carious lesions by attracting calcium and phosphate ions [54]. Self-assembling amphiphilic oligopeptide derived from cementum protein 1 (CEMP1), which is a regulator of cementum-matrix mineralization, induced intrafibrillar mineralization of collagen fibrils in the presence of calcium ions [55]. P11-4, which is another self-assembling peptide, acted as a scaffold to enhance HAP nucleation de novo [56].

### 3.3. Angiogenic Peptides

Vascularization is a crucial process during natural bone formation. Many peptides are derived from angiogenic growth factors (e.g., VEGF, fibroblast growth factor-2 (FGF-2), and PDGF), ECM (e.g., OPN, ON), and other proteins that have crucial roles in blood vessel formation [144]. 

VEGF-mimicking QK or KLT peptide (KLTWQELQLKYKGIGGG), which is derived from the VEGF receptors binding domain 17–35, not only induces EC migration and proliferation but also triggers other complex processes, like chemotaxis and capillary sprouting and organization similar to VEGF [57]. PDGF-BB-derived PBA2-1c peptide interacts with α- and β-PDGF receptors. Though its in vivo proangiogenic activity is still unclear, it functions similarly to PDGF in establishing mature blood vessels that are created by VEGF [58]. Exendin-4, which is a glucagon-like peptide 1 (Glp-1) analog, stimulates human umbilical vein endothelial cells’ (HUVECs’) motility, sprouting, and tube formation in vitro, in addition to in vivo sprout outgrowth [59]. While OPN is widely distributed in the bone matrix to help with bone metabolism, OPN-derived peptide (OPD) does not induce EC proliferation in vitro. However, like VEGF, it facilitated EC migration and tube formation using 3D collagen gels [145], suppressed osteoclastogenesis [43], and promoted the adhesion and proliferation of MSCs, as well as neovascularization in a rat tibial defect model [146]. SPARC113 and SPARC118, which are two OPN-derived peptides that exhibit potent angiogenic activity [147], stimulated in vivo angiogenesis when delivered through MMPs degradable hydrogel [60]. TP508 enhanced neoangiogenesis in femoral defects produced in rats [61] and mice [134] following one hour of local administration. The synthetic 12-mer peptides, known as RoY peptides, which were created via the phage-display technology, may also promote in vitro EC proliferation, tube formation, and sprouting, as well as induce in vivo angiogenesis via a distinct mechanism from VEGF [62]. 

## 4. Peptides for Cartilage Regeneration

### 4.1. Chondroinductive/Chondrogenic Peptides

Numerous peptides were identified to imitate the functions of ECM components, cell–cell junction molecules, and chondroinductive/chondrogenic ligands triggering specific cell-signaling pathways. Motif-derived fibronectins, like RGD, decorin, collagen, and MMPs, display chondrogenic properties. These peptides are often used to functionalize scaffolds that encourage chondrocyte adhesion, migration, and proliferation, in addition to MSC differentiation into the chondrogenic lineage.

#### 4.1.1. TGF-β Mimetic Peptides

TGF-β improves cell differentiation, collagen synthesis, and matrix deposition in cartilage tissue engineering [148]. Therefore, peptides mimicking TGF-β activity were used for cartilage tissue regeneration. TGF-β mimetic peptides, i.e., cytomodulins (CMs), are oligopeptides containing 4–6 amino acids [149]. CMs immobilized on a solid surface can potentially induce chondrogenic differentiation better compared with its soluble form (Figure 2d) [63,64].

#### 4.1.2. BMP2-Derived/Mimetic Peptides

BMP-2, which is a member of the TGF-β super-family, is one of the main chondrogenic growth factors that induce in vitro chondrogenic differentiation and cartilage regeneration in vivo. Human MSCs (hMSCs) cultured with ≥100 µg/mL of the BMP peptide (KIPKASSVPTELSAISTLYL) resulted in glycosaminoglycan (GAGs) production and increased levels of collagen production and matrix accumulation without extensive upregulation of hypertrophic markers [63,64,150,151,152]. The injection of BMP-2 mimetic CK2.1 peptide into a mouse’s tail vein enhanced chondrogenesis and articular cartilage formation without any effects on osteogenesis or BMD [65]. BMP peptide stimulated chondrogenic differentiation of hMSCs without additional growth factors. At a 100 μg/mL concentration, BMP peptide enhanced proteoglycan production and chondrogenic gene expression without causing hypertrophy, as occurs with BMP-2 [153].

## 5. Other Supporting Peptides 

### 5.1. Adhesion, Binding, or Affinity Peptides

RGD peptide: The RGD (Arg-Gly-Asp) motif is an essential cell adhesion peptide found in collagen, FN, vitronectin (VN), etc., that promotes cell adhesion and/or differentiation through binding with transmembrane integrin receptors (Figure 3a). RGD sequences, such as GRGDS, RGDS, YRGDS, and c(RGDfk), are widely used in the field of cartilage and bone repair. Bone regeneration using cyclic RGD in a sheep spinal fusion model was comparable with that of rhBMP-2 due to its increased receptor affinity [66]. RGD, particularly the longer chain RGD (GPenGRGDSPCA), was not always effective in increasing cell adhesion and spreading. This could be because pre-absorbed proteins with stronger binding sites (like FN and VN) compete with weekly adsorbed RGD for cell binding, necessitating covalent attachment [69,154]. RGD is usually capable of increasing the cellular adhesion of MSCs but is unable to induce osteogenic differentiation without osteogenic supplements or BMP-2 (Figure 3b) [109,116,117].

PHSRN peptide: PHSRN is an FN-derived peptide that synergistically enhances cellular activity with RGD and other adhesion peptides through α5β1 integrin binding, though it is ineffective alone [67,68]. Due to competitive binding with α5β1 integrin sites, RGD and PHSRN did not improve the spreading of rat bone MSC on titanium surfaces [156]. However, a dimeric platform consisting of RGD and PHSRN, with a spacing between them resembling natural fibronectin, significantly improved the spreading and proliferation of SAOS-2 osteoblast-like cells when compared with RGD alone or their combination [68]. Further improvements in attachment, spreading, and cell differentiation were made possible by a longer-chain RGD that allowed for wider spacings between the peptides like natural fibronectin; nevertheless, such modifications did not result in any osteogenic activity [157].

FHRRIKA and KRSR peptides: FHRRIKA peptide, which is isolated from the heparin-binding domain of BSP, significantly enhanced cell proliferation, spreading, and matrix mineralization [69,70]. KRSR, which is another peptide isolated from heparin-binding sites of BSP, along with FN, VN, OPN, thrombospondin, etc. [158], increased osteoblast adhesion and osteogenic gene expression [71,72,73,74]. A combination of FHRRIKA and the longer-chain RGD (CGGNGEPRGDTYRAY) stimulated rat osteoblast matrix mineralization, which was absent when they were employed alone [46]. Similarly, KRSR peptides delivered with RGD promoted cellular attachment on a glass surface due to this dual-binding mechanism with proteoglycans and integrins, respectively [158]. 

N-Cadherin is a calcium-dependent adhesion molecule consisting of five extracellular domains, in which the EC-1 domain has the His-Ala-Val (HAV) motif and plays a substantial role in cell adhesion [75]. hMSC-seeded nanofibrous hydrogels formed using a cadherin-mimic self-assembling peptide significantly upregulated the expression of chondrogenic markers, such as Col-II, Sox-9, and aggrecan [159]. hMSC-seeded hydrogels prepared from self-assembling KLD-12 peptide coupled with N-cadherin mimetic HAVDI peptide remarkably increased chondrogenesis and upregulated cartilage-specific gene expressions (collagen II, Aggrecan, and Sox9) [160]. hMSC-seeded hyaluronic acid hydrogels functionalized with RGD and N-cadherin mimetic peptide improved osteogenic gene expression and new bone formation after 12 weeks of rat MSC-seeded hydrogel implantation into the rat calvarial defects [161]. On the other hand, the NEMO-binding domain (NBD) peptide was shown to promote osteoblast differentiation and inhibit bone resorption [76] and osteoclastogenesis [77]. The other integrin-binding peptides derived from collagen IV (NYYSNS), fibronectin (KQAGDV, PRARI), laminin (IKLLI, LRGDN, and SINNNR), fibrinogen (GWTVFQKRLDGS, YSMKKTTMKIIPFNRLTIG, and GWTVIQNRQ), netrin-1 (QWRDTWARRLRKFQREKKGKCRKA), CAM-1 (QIDS, LDT, and DELPQLVTLPHPNLHGPEILDVPST), thrombospondin (CSVTCG), and nidogen-1 (FRGDGQ) have been reported [162].

Affinity peptides can bind with specific cells, scaffolds, and cytokines associated with cartilage regeneration. Using an anti-CD44 antibody, biotin–avidin binding system [163], chondrocyte affinity peptides [164], E7 peptide [165], etc., cell adhesion was greatly improved. A TGF-β affinity peptide that can recruit TGF-β to the impaired region was ligated to the PA nanofibers and promoted in vitro chondrogenic differentiation of hMSCs and cartilage repair in rabbits [166,167]. ECM affinity peptides mimicking the native environment of chondrocytes were integrated with scaffolds [168,169]. Hydrogels modified with HA and chondroitin sulfate-binding peptides significantly promoted BMSC differentiation into a chondrogenic lineage (Figure 3c) [86].

While RGD peptides immobilized at a low density effectively promoted cell adhesion and chondrogenesis [170,171,172], a higher density of RGD induced the hypertrophic transformation of chondrocytes due to the activation of the integrin-mediated pathway [173,174,175]. However, compared with simple RGD adsorption, the chemical grafting of RGD peptides onto the biomaterial surface offered improved coating stability and osteoconductivity (Figure 3d) [155]. The incorporation of the N-cadherin-mimetic HAVDIGGGC peptide into HA hydrogels increased chondrogenic differentiation [176,177]. Similarly, laminin-related CDPGYIGSR peptide significantly promoted the adhesion of bovine knee chondrocytes onto the polyethylene oxide PEO/chitosan scaffolds [78].

### 5.2. Cell-Penetrating Peptides (CPPs)

CPPs are peptides derived from bacteria, viruses, or a synthetic method transverse the cellular membrane to transport their ‘cargo’ (proteins, siRNA, nanoparticles, oligonucleotides, and other peptides) into the cytoplasm [152]. hMSCs delivered with CPP-conjugated co-activator-associated arginine methyltransferase 1 (CARM1) proteins, transcriptional factor fusion proteins, or recombinant adenovirus expressing BMP-2 promoted in vitro osteogenic differentiation and in vivo bone formation [178,179,180]. NLS-TAT peptides facilitated the introduction of the hTGF-β3 plasmid into self-assembled peptide scaffolds or pre-cartilaginous stem cells (PCSCs) directing chondrogenic differentiation [83,84]. 

### 5.3. Peptides Promoting Cell Migration

Migration of fibroblasts, MSCs, osteoblasts, chondrocytes, and EC toward the defect site is required for OCD healing. ECM proteins and GFs (EGF, PDGF) act as mitogens or chemotactic factors facilitating cell migration [181]. Peptides derived from them often exhibit superior cell migration than the parent protein. Stromal cell-derived factor-1α (SDF-1α) is a known recruiter of endothelial progenitor cells during osteochondral repair [182]. SDF-1-derived elastin-like peptide (SDF1-ELP) showed comparable EC migration and vascularization to SDF and even more effective healing of full-thickness diabetic wounds in mice [79]. Histatin-1, which is a 38-mer antimicrobial peptide found in saliva, exerts a proangiogenic effect that stimulates the adhesion and migration of EC [80]. The endogenous histatins, annexin A1-derived peptide Ac2-26 [81], or frog-skin-derived esculentin 1–21 peptide [82] showed better wound-healing activity driven by cell migration. The administration of LL-37 significantly increased vascularization because of enhanced EC migration [183]. Like FGF-2 (bFGF), the FGF-2-mimicking peptide YRSRKYSSWYVALKR derived from the 106–120 peptide domain in FGF-2, which is a partial agonist of FGF receptors, could successfully promote in vitro HUVEC cells proliferation and migration [184]. 

### 5.4. Self-Assembly (SA) Peptides

SA peptides are composed of either alternating hydrophilic and hydrophobic amino acids or peptide amphiphiles (PA), which can self-assemble into β-sheet structures and interwoven nanofibrous hydrogel matrices under physiological conditions [185,186,187]. SA peptides isolated from various growth factors and bone-related proteins (RANKL-binding peptide, AC-100, B2A) were studied for osteoblast differentiation and bone regeneration [21,188,189]. A RADA16-I peptide immobilized onto the BMP-2 loaded hydrogel promoted osteogenic differentiation of MSCs, leading to a higher expression of osteogenic-related genes (ALP, OCN, Runx_2_) and in vivo bone regeneration [85]. PuraMatrix^TM^, which is a commercially available RADA16-1-peptide-containing hydrogel, strongly supported cartilage formation [190,191]. Repeating units of KLD and RAD in a peptide hydrogel enhanced cartilage formation compared with an agarose gel [192].

### 5.5. Degradable Peptides

Degradable peptides accelerate the degradation of scaffolds to facilitate cell penetration and migration, ECM synthesis, and deposition, and consequently, tissue ingrowth. MMPs are the best examples of ECM-degrading proteins. The inclusion of an MMP-derived peptide (KCGPQGIWGQCK) remarkably promoted GAGs and collagen deposition onto crosslinked poly(ethylene glycol) norbornene hydrogels [86,87].

### 5.6. Antimicrobial and Immunomodulatory Peptides

AMPs or cationic host defense peptides (CHDPs) are oligopeptides (5–100 amino acids); cationic (net charge: +2 to +13), hydrophobic (50% hydrophobic residues), or amphipathic; derived from bacteriophages, bacteria, fungus, plants, and animals; and possess broad-spectrum antibacterial properties without causing antibiotic resistance [193]. Few also have angiogenic, cell migratory, and immunomodulatory properties that facilitate tissue regeneration. 

Helix-based (LL-37, magainin, melittin); sheet-based (protegrins, bactenecin, defensins); coil-based (indolicidin, omiganan); and composite AMPs, such as melimine, have been reported [88]. LL-37, which is a 37-mer peptide α-helical AMP derived from human cathelicidin antimicrobial protein, possesses antibacterial, antibiofilm, immunomodulation, and angiogenesis properties. Pexiganan, which is a synthetic magainin analog, and PLG0206 (RRWVRRVRRWVRRVVRVVRRWVRR), which is rich in arginine, valine, and tryptophan residues, reduce biofilm formation. Protegrin-1, bactenecin and its derivatives (RKWRIVVIRVRR and RWRRIVVIRVRR), and the arginine-rich PEP8R peptide show broad-spectrum antimicrobial activities and reduced cytotoxicity. Indolicidin, which is rich in tryptophan and proline, is highly potent against bacteria, fungi, and viruses, whereas omiganan shows improved bioactivity with reduced cytotoxicity. 

AMPs work either by disrupting the bacterial cell wall or cytoplasmic membrane (Figure 4a) or by interfering with cell wall synthesis. The bactericidal properties of keratin-derived anti-inflammatory KAMP-19 peptide originate from the destruction of the bacterial cell membrane via pore formation [90]. α-defensin, which is known as human Neutrophil Peptide 1 (HNP-1), through its high affinity for lipid II, destabilizes the cell wall integrity [91]. Likewise, RWRWRW-NH_2_ delocalizes peripheral membrane proteins [92], LL-37 affects the Sfp1 gene associated with cell wall synthesis, and buforin II and TO17 interfere with nucleic acid functions [89,194].

Biofilms associated with chronic bone infection are difficult to treat because of bacterial resistance [197]. LL-37 interacts with the cell wall component mannan to lower Candida adhesion (Figure 4b) [195]. AMP 1037, which is a 9-mer peptide, decreases the swimming and motility of bacteria [93]. Piscidin and sculentin (1–21) degrade a pre-formed biofilm matrix by activating nuclease activity and disturbing the membrane function, respectively [94,95].

Anti-inflammatory cytokines (AMPs) can suppress pro-inflammatory cytokine signaling pathways or stimulate the synthesis of anti-inflammatory cytokines associated with an innate and adaptive immune response (Figure 4c). Immunomodulatory metalloproteinases (AMPs), such as cathelicidin-WA, stimulate macrophage polarization from the M1 phenotype (caused by *E. coli*) to the M2 phenotype (facilitating bone repair) (Figure 4d) [196]. Macrophage polarization plays a significant role in mediating chondrogenesis. Failure in the M1 to M2 transformation may lead to the progression of cartilage injury and interrupted tissue remodeling and repair [198,199]. 

Regenerative AMPs are beneficial for the regeneration of bone tissue because they can inhibit osteoclastogenesis and stimulate angiogenesis. Histatin-1, which is a 38-mer peptide occurring in saliva, induces angiogenesis and tube formation [80]. On the other hand, human β defensin 3 stimulates MSC osteogenesis [96]; GL13K (GKIIKLKASLKLL) suppresses osteoclastogenesis [97]; and LL-37 supports the proliferation, migration, and differentiation of MSCs to osteogenic lineage [200]. 

## 6. Peptide-Conjugated Biomaterials

Various peptides with osteo- or chondrogenic properties were delivered or conjugated with synthetic and natural biomaterials for better OCD healing.

### 6.1. Osteo-Inductive Scaffold 

Delivery of the BMP2-derived KIPKASSVPTELSAISTLYL peptide through TCP scaffolds promoted bone healing in rabbit radial bone defects [201]. BMP-2 mimetic peptides attached to a substrate enhanced the BMSC attachment and differentiation without affecting the mineralization [117]. However, when these peptides were combined with RGD, the rate of mineralization increased because RGD induced higher interaction with cells in the absence of osteogenic supplementation [116]. The incorporation of BMP-2-mimetic peptide conjugated to a heptaglutamate moiety (E7-BMP-2) into the mineralized PLGA-collagen-gelatin nanofiber promoted bone formation in critical-sized alveolar bone defects in rats after 4 weeks of implantation [202]. An electrospun porous cellulose acetate nanofibrous mat modified with adhesive peptides KRSR, RGD, and growth factor BMP-2 enhanced the adhesion and proliferation of pre-osteoblastic cells [203]. PCL nanofibrous scaffolds containing different concentrations of the bioactive with the KIPKASSVPTELSAISTLYL peptide derived from the BMP-2 scaffolds developed by Lukasova et al. and seeded with porcine MSCs showed an increased expression of OCN, and collagen I significantly increased osteogenic differentiation [204]. KRSR, BMP mimetic peptides, and FHRIKKA were successfully integrated into the outer, middle, and inner layers of PLGA electrospun membranes through layer-by-layer assembly, and such modification demonstrated synergistic effects on bone healing by enhancing cell attachment, differentiation, and mineralization [205]. FHRRIKA and KRSR peptides that were covalently linked with the surface of polymeric nanoparticles significantly improved MSC attachment and ALP activity due to osteogenic differentiation, which was not very profound in the case of KRSR [206,207]. The immobilization of KRSR on a silane-functionalized borosilicate glass [158], micropatterning with KRSR on borosilicate glass [208], or another substrate [69,73,74,209] selectively enhanced osteoblast adhesion through αvβ5 integrin receptors binding.

On the other hand, GFOGER-functionalized PEG-BMP-2 hydrogels increased in vitro cell spreading and differentiation of BMSCs into an osteogenic lineage, as well as improved bone healing in a mice radial defect model, even in the absence of BMP-2 [109]. Implants coated with GFOGER were found to improve peri-implant bone regeneration and osseointegration [210]. However, hydroxyapatite disks adsorbed with GFOGER failed to attach cells with a decrease in cell spreading [154], possibly due to a lack of nature-mimicking conformation of GFOGER [108]. P15-coated bovine bone grafts allowed for the faster recovery of periodontal defects in humans [211]. P-15-containing bone graft substitutes could facilitate the process of early bone formation [212,213], bone healing, and regeneration [214]. P24 delivery to the defect site was achieved by encapsulating in chitosan microspheres [118], covalent binding with polymeric backbone [22], or simple adsorption on hydroxyapatite [22], which greatly enhanced the bioactive properties of the scaffolds. 

DGEA immobilized on HA was found to enhance the adhesion and osteoblastic differentiation of MSCs and new bone formation [215]. BCSP™-1, when delivered using HA and tricalcium phosphate (TCP) grafts, stimulated ALP activity in murine calvarial osteoprogenitor cells [216]. OGP-rich PLGA scaffold accelerated bone healing in 1.5 cm rabbit segmental defects [217]. A PEG hydrogel containing PTH1–34 promoted in situ bone augmentation in rabbits [218], whereas TP508-loaded PPF composite and microsphere scaffolds showed enhanced bone formation in rabbit segmental bone defects [219]. RGD-coated implants were found to have an increased peri-implant bone formation and enhanced direct bone apposition, even in areas of poor surrounding bone [155,220,221,222]. An increased osteoblast density was observed when an anodized nanotubular titanium surface was coated with RGD or RGDS (Arg-Gly-Asp-Ser) [223,224]. 

### 6.2. Chondro-Inductive Scaffolds

Collagen-type-II-derived peptide-conjugated zwitterionic carbon nano-dots (pCDs) induced the chondrogenic differentiation of ADMSCs. The pluronic-F-127-hydrogel-mediated delivery of pCD could successfully promote rabbit auricular cartilage defect healing after 60 days [89]. Self-assembled KLD12 and KLD12-CMP7 peptide hydrogels loaded into poly(L-lactide-co-caprolactone) scaffolds could successfully create a chondrogenic microenvironment for rat BMSC-seeded scaffolds after subcutaneous implantation in nude mice [225]. Self-assembling PA scaffolds loaded with a TGF-binding domain, namely, HSNGLPLGGGSEEEAAAVVV(K)-CO(CH_2_)_10_CH_3_, could support the viability and chondrogenic differentiation of hMSC, as well as accelerate the healing of articular cartilage defects in rabbit [167]. Similarly, HSNGLPL-peptide-coated porous chitosan scaffolds or gelatin methacryloyl (GelMA) hydrogel successfully induced the chondrogenic differentiation of MSCs and recovery of osteochondral defects in mice and rabbit models [226,227]. RGD coating on the synthetic bioinert materials surface significantly improved properties like cell adhesion, viability, and differentiation. Composite scaffolds were prepared from an RGD-functionalized hydroxyapatite/methoxy poly(ethylene glycol)-block-poly(ε-caprolactone) (1:2 ratio) scaffold followed by infiltration of TGF-β1 functionalized glycidyl methacrylate-hyaluronic acid hydrogel. The scaffolds significantly upregulated the expression of cartilage-specific genes (aggrecan, Col2a1, Sox9) with higher accumulation of sulfated glycosaminoglycan (sGAG), whereas successful OCD healing was also achieved after 12 weeks of implantation in a rabbit knee defect model [228]. 

### 6.3. Multifunctional Scaffolds

The subcutaneous injection of nanofibrous hollow microspheres conjugated with TGF-β1 mimetic CM10 combined with rabbit BMSCs successfully induced ectopic cartilage formation in nude mice via the chondrogenic differentiation of MSCs; in contrast, BMP-2 mimetic P24-peptide-loaded microspheres demonstrated osteogenic activity leading to bone formation [64]. The implantation of 3D printed porous bi-layered scaffolds infiltrated with TGF-β1 binding HSNGLPLGG(MA)-peptide-rich GelMA hydrogels in the top layer and hydroxyapatite in the bottom layer successfully repaired the osteochondral defects in SD rats through simultaneous cartilage and bone regeneration, respectively [229]. Hyaluronic-acid-based hydrogel particles (HGPs) functionalized with a cysteine-tagged CK2.1 peptide showed promising results for cartilage repair in a mouse model without inducing chondrocyte hypertrophy, unlike BMP-2 [28]. Meanwhile, using the same model, CK2.2 and CK2.3 peptides induced osteoblast differentiation and mineralization, similar to BMP-2 [65]. RGD-functionalized poly(ethylene glycol)-diacrylate (PEGDA) hydrogel encapsulating rat osteoblasts facilitated the adhesion and spreading of rat osteoblasts, leading to matrix mineralization [230]. RGD immobilized on hydroxyapatite improved the healing of femoral condylar defects in rabbits [231]. However, a longer chain RGD was unable to do the same without a TGF-β supplementation [232]. RGD peptide attached to glass coverslips through silanization did not show osteoblast adhesion since attachment happened through integrin binding [158]. 

Wang et al. developed a cryogenic 3D printed bilayer β-tricalcium phosphate/PLGA osteochondral scaffold that was enriched with osteogenic peptide sequence, KIPKA SSVPT ELSAI STLYL SGGC, and TGF-β1-loaded collagen I hydrogel for cartilage repair. The osteochondral scaffold showed improved chondrogenic differentiation of rBMSCs via the upregulated expression of chondrogenic markers and glycosaminoglycan (GAG) production, while it also enhanced the osteogenic differentiation of rBMSCs at the subchondral layer via the upregulated expression of RUNX2, osteocalcin and alkaline phosphatase (ALP), and calcium deposition [233].

Due to the bactericidal or anti-biofilm, immunomodulatory, and regenerative capacity, AMPs with adequate cytocompatibility have been explored for bone and cartilage tissue regeneration. A titanium surface immobilized with Mel4 peptides demonstrated significant antimicrobial efficacy after implantation into the rabbit femoral defects [234]. While a GL13K peptide (GKIIKLKASLKLL-NH_2_)-coated titanium surface could effectively inhibit peri-implantitis through immunomodulatory function [235,236], a titanium plate modified with antimicrobial LF1-11-coupled RGD peptide demonstrated good antibacterial properties with improved osteoblast adhesion, proliferation, and mineralization [237]. The inclusion of HHC-36 (KRWWKWWRR), which is a broad-spectrum cationic AMP, and osteoconductive Laponite nanosilicates into GelMA hydrogel, which was used for coating of a titanium implant surface, significantly promoted the expression of osteogenic-related genes and mineralization of hMSCs, as well as effectively inhibited Gram-positive bacteria (*S. aureus* and *S. epidermidis*) and Gram-negative bacteria (*P. aeruginosa* and *E. coli*) [238]. On the other hand, HA bone scaffolds coated with PSI10 (RRWPWWPWRR) and BMP2-MP [239] or a nano-hydroxyapatite coated titanium surface containing antimicrobial human β-defensin 3 (HBD-3) peptide and BMP-2 [240] significantly promoted the antibacterial and osteogenic activities of the scaffolds. 

Multifunctional PEEK coated with lithium-ion and mussel-inspired antimicrobial peptide enhanced the osteogenesis-associated genes/proteins expression and osseointegration at the bone–implant interface, as well as the inhibition of *E. coli* and *S. aureus* [241]. A coating with mouse beta-defensin-14 (MBD-14) also improved the antimicrobial and osseointegration properties of PEEK [242]. A KR-12 analog immobilized on the surface of titanium or PEEK bone implants via a polydopamine coating showed remarkable antibacterial activity, osteogenic differentiation of MSCs, and peri-implant bone formation [243,244]. 

## 7. Summary and Outlook

Several bioactive peptides were identified in this review based on their ability to support the healing of cartilage and bones. Precisely, peptides derived from collagen (GFOGER, P15, KOD, BCSP™-1) and BMPs (P24, BFP-2, CK2.2, and CK2.3) can induce osteogenic differentiation, whereas NCP-derived peptides (8DSS, 3NSS, shADP5, QP5, P26, P32, LRAP, TDP, CEMP1) facilitate biomineralization activity during bone formation. Supplementation with angiogenic QK, TP508, or RoY peptides may eventually form matured bone. On the other hand, CK2.1, CM, or peptides mimicking TGF-β and BMP-2 activity can facilitate cartilage tissue formation. 

The adjuvant peptides (RGD, PHSRN, SDF1-ELP, NLS-TAT, RADA16-I, KCGPQGIWGQCK) may support by facilitating cell adhesion, migration, intracellular cargo delivery, or the creation of a 3D microenvironment through self-assembly. Peptides such as LL-37, GL13K, KAMP-19, AMP 1037, and β defensin-3 can treat post-implantation issues like infection, inflammation, or an immunogenic reaction. In this context, pattern-based searching using artificial intelligence (AI) and machine learning (ML) tools can identify more unique sequences or bioactive motifs from full-length proteins. Considering the structural and functional heterogeneities of OC units, a strategy involving multiple bioactive peptides or combining them with biomaterials for simultaneous bone and cartilage regeneration could be fruitful. For instance, the blending of collagen-mimetic GFOGER or P15 peptides, or BMP-2 mimetic P24 or BFP-2 peptides, along with VEGF-mimetic QK, TP508, or RoY peptides, can synergistically promote osteo-inductive effects. 

Scaffolds or hydrogels and other bone graft substitutes coupled with GFOGER, FHRIKKA, KRSR, P-15, DGEA, or adhesive peptides (RGD or PHSRN) may significantly improve osteogenic activity and in vivo bone formation. The delivery of HSNGLPL, KLD12, and KLD12-CMP7 peptides through pluronic F-127 hydrogel, chitosan, or poly(L-lactide-co-caprolactone) scaffolds may promote cartilage regeneration. Short SA peptides co-assembled with macromolecules may enhance the mechanical and biological properties of hydrogel nanocomposites. Cationic AMPs, such as GL13K or HHC-36, were often valuable for enhanced osteogenic activity apart from their inherent antimicrobial or immunomodulatory properties.

Multifunctional GelMA/HA, PEGDA, β-TCP/PLGA scaffolds or hydrogels, PEEK, and titanium implants combined with osteo- and chondrogenic peptides may facilitate the simultaneous healing of OCDs. However, more study on electrospinning, inkjet printing, and bioprinting is required to accomplish the spatial distribution of multifunctional peptides within the constructs that resemble the intricate geometry and multi-scale hierarchy of native tissue.

## Figures and Tables

**Figure 1 biomedicines-12-00313-f001:**
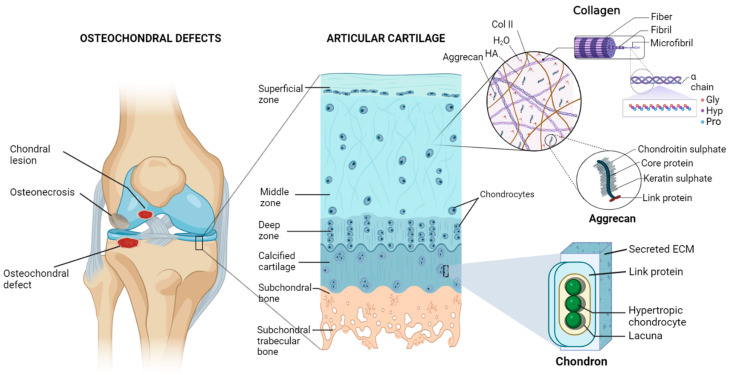
Diagram of the osteochondral unit showing the structural hierarchy, zonal arrangements, and extracellular matrix composition of the articular cartilage. The cartilage matrix is mainly composed of collagen type II (Col II), hyaluronic acid (HA), aggrecan, and a large amount of water (H_2_O). Alignments of chondrocytes are flattened, spherical, and columnar in superficial, middle, and deep zones, respectively. On the other hand, chondrocytes are hypertrophic in the calcified matrix, primarily made of nanohydroxyapatite and collagen type I, and found inside lacunae. Pro—proline, Hyp—hydroxyproline, Gly—glycine. Figure generated via biorender.com.

**Figure 2 biomedicines-12-00313-f002:**
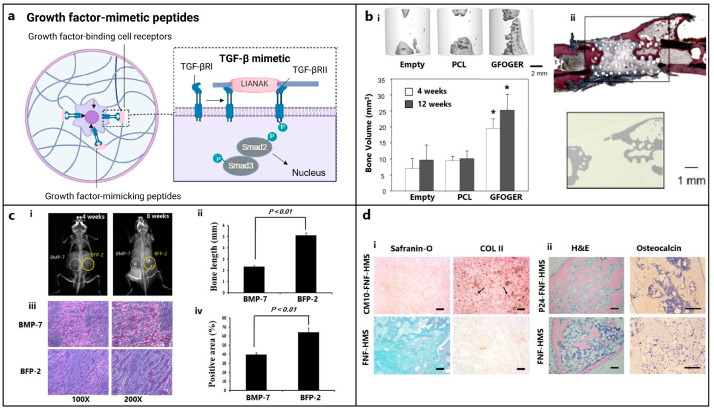
Growth-factor-mimetic peptides for bone and cartilage regeneration in osteochondral defects. (**a**) Growth-factor-mimetic peptide epitopes inducing direct signaling, e.g., mimicking transforming growth factor-β (TGF-β) pathway for TGF-β receptors (TGFβR) binding (figure generated via biorender.com). (**b**) (**i**) Critical bone defects treated with GFOGER-coated scaffolds show significantly higher bone formation compared with uncoated PCL scaffolds and empty defect (control), as revealed using micro-CT imaging after 12 weeks of implantation; the plot also depicts a similar pattern at 4 weeks. (* Different from empty defect and PCL (*p* < 0.05)) (**ii**) Histology confirms that areas of high attenuation in GFOGER-coated sample, as revealed using 2D micro-CT image (inset) are bone tissue (red/pink) (soft tissue appearing blue/green in Sanderson’s rapid bone stain) [108]. Copyright© 2009 Elsevier Ltd. (**c**) Bone formation in mice treated with BFP-2 for 4 or 8 weeks in comparison with BMP-7 was evaluated using (**i**) radiography, (**ii**) bone length estimation, (**iii**) histological assessment after hematoxylin and eosin staining, and (**iv**) measurement of the osteogenic area from histology samples using the Image-Pro Plus 6.0 software (*n* = 4) [27]. Copyright© 2017, Springer Nature. (**d**) Histological analysis of (**i**) CM10-conjugated FNF-HMS (CM10-FNF-HMS) with rabbit BMSCs demonstrates positive SO staining (GAGs) and collagen type II (COL II) staining, indicating hyaline cartilage formation, after 2 weeks (arrows indicate collagen deposition) and (**ii**) P24-conjugated FNF-HMS (P24-FNF-HMS) with rabbit BMSCs reveals significantly higher bone formation in H&E and osteocalcin staining after 5 weeks of subcutaneous implantation compared with FNF-HMS (D–F) (scale bars: 100 μm) [64]. Copyright© 2014 WILEY-VCH Verlag GmbH & Co., KGaA, Weinheim.

**Figure 3 biomedicines-12-00313-f003:**
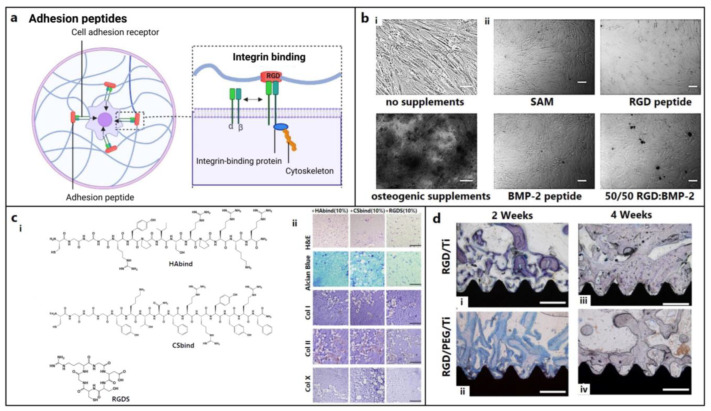
Role of other supporting peptides for osteochondral regeneration. (**a**) Signaling of the adhesion peptides, e.g., integrin binding (figure generated via biorender.com). (**b**) Evaluation of mineralization activity at 3 weeks of hBMSC culture on (**i**) tissue culture plate with or without osteogenic supplements and (**ii**) self-assembled monolayer (SAM), bone morphogenic protein (BMP), and RGD peptide gradients on glass coverslips supplemented with osteogenic medium. Scale bars: 100 μm [116]. Copyright© 2011 Elsevier Ltd. (**c**) (**i**) Structures of acrylated streptococcal collagen-like 2 (Scl2) protein conjugated with hyaluronic acid (HA) binding CGGGRYPISRPRKR peptide (HAbind), chondroitin sulfate (CS) binding CGGGYKTNFRRYYRF peptide (CSbind), and cell adhesive GRGDSC peptide (RGDS); (**ii**) HAbind, CSbind, and RGDS peptide-based hydrogels cultured with hMSCs for 4 weeks showed uniform cell and ECM distribution (H&E staining in panel 1); extensive sGAG accumulation (Alcian Blue staining in panel 2); and high, low, and negative expressions of collagen type I (cartilage marker, in panel 3), type II (bone marker, in panel 4), and type X (hypertrophic marker, in panel 5), respectively (IHC staining), in all hydrogels, indicating chondrogenic differentiation without any hypertrophic transformation. Scale bars: 200 μm [86]. Copyright© 2015 Elsevier Ltd. (**d**) Histological analysis showing higher bone formation on RGD/PEG/Ti implants (**ii**,**iv**) compared with RGD/Ti implants (**i**,**iii**) after 2 and 4 weeks of implantation in rabbit femoral condyles, respectively. Mag: 40×; scale bar: 0.5 mm [155]. Copyright© 2011 Elsevier Ltd.

**Figure 4 biomedicines-12-00313-f004:**
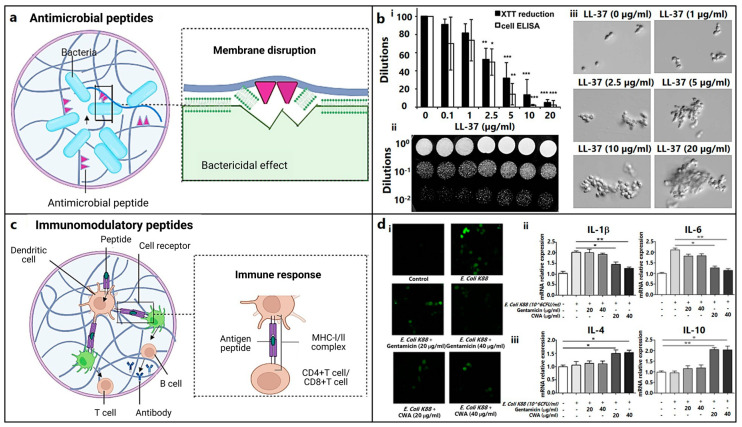
Role of antimicrobial and immunomodulatory peptides for osteochondral regeneration. (**a**) Schematic representation of bacterial cell membrane damage using antimicrobial peptides (figure generated via biorender.com). (**b**) Inhibition of *C. albicans* adhesion with different concentrations of LL-37 evaluated using (**i**) XTT reduction assay and whole-cell ELISA (*, *p* < 0.05; **, *p* < 0.01; ***, *p* < 0.001), (**ii**) spot assay, and (**iii**) optical microscopy of the floating cells (mag: 400×) [195]. Copyright© 2011 Tsai et al. (**c**) Mechanism of immunomodulatory peptides eliciting immune-cell response (figure generated via biorender.com). (**d**) *E. coli* K88-induced RAW264.7 cells (macrophages) treated with varying doses of cathelicidin-WA (CWA) and gentamicin shows (**i**) significant reduction in reactive oxygen species (ROS) (in fluorescent images), (**ii**) decrease in proinflammatory cytokine (IL-1β, IL-6), and (**iii**) increase in anti-inflammatory cytokine (IL-4, IL-10) levels [196]. Copyright© 2017 Elsevier B.V. (* indicates *p* < 0.05 and ** indicates *p* < 0.01).

**Table 1 biomedicines-12-00313-t001:** Bioactive peptides for osteochondral regeneration.

Types	Source Molecule	Mode of Delivery	In Vitro/In Vivo Activity	[Refs]
**a. Osteogenic peptides**				
(i) Osteo-inductive peptides				
GFOGER	Col	Ti implant	Osteogenic differentiation of BMSCs, mineralization in rat	[14]
GTPGQGIAGQRGVV (P15 peptide)		HAP scaffold	↑ Osteogenic effect in rat	[15]
((PKG)_4_-(POG)_4_-(DOG)_4_) (KOD peptide)		Peptide scaffold	↑ GAG and Col deposition in rabbit	[16]
DGEA		PA scaffold	Osteogenic differentiation of hMSC	[17]
NGLPGPIGP (BCSP™-1)		Injection	In vitro mineralization of bone marrow-derived osteoblasts and in rats	[18]
GPAGPHGPVG, APDPFRMY, TPERYY		Culture medium	↑ Osteogenic activity in MC3T3-E1 cells	[19]
KIPKASSVPTELSAISTLYL	BMP-2	Culture medium, TCP scaffold	↑ ALP activity in murine osteoprogenitor cells and bone formation in rabbit	[20,21]
SKIPKASSVPTGLSAISTLYLAAA (P24)		PLGA/PEG-ASP scaffold	Bone formation in Wistar rats	[22,23,24]
CKIPKPSSVP-TELSAISMLYL (PEP7)		Ti implant	Bone formation in human osteoblast-like cell	[25]
KIPKASSVPTELSAISTLYL		Alginate gel	Osteogenic differentiation of MSC, C3H10T1/2 and bone formation in rat	[20]
NSVNSKIPKACCVPTELSAI, KIPKASSVPTELSAISTLYL, DWIVA		Alginate gel	↑ Osteogenic activity in C3H10T1/2 cells and bone formation in rat	[11]
RKKNPNCRRH	BMP-4	Alginate gel	↑ Osteogenic activity in hMSCs and bone formation in rabbit	[26]
VEHDKEFFHPRYHH (BFP-2)	BMP-7	BFP-2-treated BMSCs	↑ Osteogenic activity in BMSCs and ectopic bone formation in mice	[27]
TVPKPSSAPTQLNAISTLYF, GQGFSYPYKAVFSTQ, ETLDGQSINPKLAGL		PET sheet	↑ Osteogenic activity of BMSCs and bone formation in mice	[26]
KVGKACCVPTKLSPISVLY	BMP-9	PET sheet	↑ Osteogenic activity and in vivo bone formation	[26]
CK2.2, CK2.3	Synthetic	Tail vein injection	↑ BMP signaling in C3H10T1/2 cells and mice model	[28]
PTHrP1–34, PTHrP1–36, PTHrP107–111	PTH	s.c. injection	Osteogenic differentiation of BMSCs, bone formation in ovariectomized mice	[29,30,31]
CGRP–α and β-CGRP	CGRP	Culture medium	Osteogenic effects in hOBs	[32,33,34,35]
ALKRQGRTLYGFGG (OGP)	Mammalian blood	Culture medium	↑ Osteogenic effects in BMSCs	[36,37,38]
AGYKPDEGKRGDACEGDSGGPFV (TP508)	Thrombin	Culture medium	↑ Proliferation, osteogenic differentiation (osteoblast)	[39,40]
FN III9-10/12-14	FN	Coating on Petri plate	↑ Osteoblast activity in human osteoblasts	[41]
CBM	OPN	Collagen scaffold	Osteogenic differentiation of BMSCs and bone formation in rabbit	[42]
SVVYGLR	OPN	Gelatin–PEG Tyr hydrogel	↑ Neovascularization in HUVEC and s.c. injection in mice	[43,44,45]
FHRRIKA	BSP	Quartz surface	↑ Osteoblast activity in osteoblast-like cells	[46]
(ii) Biomineralization peptides				
8DSS	DPP	Soaking in solution	↑ Remineralization of enamel ex vivo	[47,48]
3NSS	DSS	Soaking in solution	↑ Remineralization of human enamel ex vivo	[49]
DSESSEEDR	DMP-1	Soaking in solution	↑ Mineralization of demineralized dentin	[50]
shADP5, QP5, P26, P32	Amelogenin	Soaking in solution	↑ Remineralization of the dentin ex vivo	[51,52]
LRAP	Amelogenin	Chitosan hydrogel	↑ Mineralization of human tooth ex vivo	[53]
TDP (DRNLGDSLHRQEI)	Tuftelin	Soaking in solution	Remineralization of enamel caries ex vivo	[54]
NNCCCCRRES(p)	CEMP1	Soaking in solution	↑ Remineralization of enamel	[55]
P11-4 (Ac-QQRFEWEFEQQ-NH_2_)	-	Hydrogel	↑ Remineralization of dentinal collagen	[56]
(iii) Angiogenic peptides				
QK	VEGF	Gelatin-coated dish	↑ Migration and proliferation of EC	[57]
PBA2-1c	PDGF-BB	PLG scaffold	Angiogenesis in mice	[58]
Exendin-4	Exendin-4	s.c. injection of matrigel plug	↑ Tube formation in HUVECs and angiogenesis in mice	[59]
SPARC113, SPARC118	OPN	PEG hydrogel	↑ Angiogenesis in mouse and rat (s.c. injection)	[60]
TP508	Thrombin	Percutaneous injection	↑ Neo-angiogenesis in rat bone fracture	[61]
RoY	Synthetic	Culture medium; s.c. injection	↑ Proliferation, sprouting of HUVEC, in vivo angiogenesis in mice	[62]
**b. Chondroinductive peptides**				
CMs	TGF-β	PHEMA-g-PLLA-acrylic microsphere	↑ Chondrogenic differentiation in BMSCs	[63,64]
CK2.1	Synthetic	HA hydrogel	↑ Chondrogenesis in mice	[65]
**c. Other supporting peptides**				
(i) Adhesion, binding, affinity peptides				
Cyclic RGD	Col, FN, VN	Ti cage	↑ Spinal fusion in sheep	[66]
PHSRN	FN	Ti implant	↑ Adhesion of osteoblast-like cells	[67,68]
FHRRIKA	BSP	Ti implant	↑ Mineralization in osteoblast and rat model	[69,70]
KRSR	BSP, FN, VN, OPN	Calcium aluminate scaffolds	↑ Adhesion, osteogenic gene expression in mouse C3T3 fibroblasts and osteoblasts	[71,72,73,74]
HAV	N-Cadherin	PS-PEO surface	↑ Adhesion of hMSCs	[75]
NEMO-binding domain (NBD) peptide	Synthetic	Culture medium	↑ Osteoblast differentiation in C2C12 cells	[76,77]
CDPGYIGSR	Laminin	PEO/chitosan scaffold	↑ Adhesion and ECM deposition in bovine knee chondrocytes (BKCs)	[78]
(ii) Peptides supporting cell migration				
SDF1-ELP	SDF-1	Self-assembled nanoparticle	EC migration, vascularization in diabetic mice	[79]
Histatin-1	Saliva	In vitro	Adhesion, migration, and angiogenesis of EC	[80]
Ac2-26	Annexin A1	Injection	Cell migration in diabetic mice	[81]
Esculentin 1-21	Frog skin	Culture medium	Migration of HaCaT cells	[82]
(ii) Cell penetrating peptides (CPPs)				
NLS-TAT	HIV-1 Tat protein	PLGA-PLL scaffold	↑ Chondrogenic differentiation in PSC	[83,84]
(iii) Self-assembly (SA) peptides				
RADA16-I	Synthetic	Ti cylinder	↑ Mineralization in rabbit	[85]
(iv) Degradable peptides				
KCGPQGIWGQCK	MMP-derived	PEG hydrogel	↑ GAGs and collagen deposition in MSCs	[86,87]
(v) Antimicrobial and immunomodulatory peptides				
Melamine	Synthetic	In vitro	Destabilization of cell membrane in vivo	[88]
LL-37	Human cathelicidin	In vitro	Inhibits cell wall synthesis	[89]
Pexiganan	Synthetic analog of magainin-2	In vivo	Disrupts bacterial membrane in diabetic foot ulcer patient	[88]
Magainin-1	Frog skin	In vitro	Destabilizes or disrupts bacterial membrane	[88]
PLG0206	Synthetic	In vivo	Disrupts cell membrane	[88]
Protegrin-1	Porcine leukocytes	In vitro	Disrupts cell membrane	[88]
Bactenecin	Bovine neutrophils	In vitro	Inhibits protein synthesis	[88]
PEP8R	Synthetic	Hydrogel	Disrupts bacterial cell membrane	[88]
Indolicidin	Bovine neutrophil	In vitro	Increase cell membrane permeability	[88]
ILRWPWWPWRRK (Omiganan)	Synthetic	Topical application	Destabilizes cell membrane in vivo	[88]
RAIGGGLSSVGGGSSTIKY (KAMP-19)	Keratin	In vitro	Pore formation in bacterial cell membrane	[90]
HNP-1	Human neutrophil	In vitro	Destabilizes cell wall integrity	[91]
RWRWRW-NH_2_	Synthetic	In vitro	Delocalizes peripheral membrane proteins	[92]
AMP 1037	Synthetic	In vitro	↓ Swimming and motility of bacteria, gene expression	[93]
Piscidin and sculentin 1–21	Fish	In vitro	Degrades biofilm via DNA damage in vitro	[94,95]
β defensin 3	Human	Strontium titanate nanotubes	Antibacterial activity	[96]
GKIIKLKASLKLL (GL13K)	Salivary protein	Ti implant surface	Delocalization of bacterial cell membrane	[97]

Abbreviations—BMP: Bone morphogenic protein, ALP: Alkaline Phosphatase, Col: Collagen, CK: Calmodulin Complex, PTH: Parathyroid hormone, CGRP: Calcitonin gene-related peptide, OGP: Osteogenic growth peptide, FN: Fibronectin, VN: Vironectin, OPN: Osteopontin, CBM: Collagen-binding motif, BSP: bone sialoprotein, VEGF: Vascular endothelial growth factor, PDGF-BB: Platelet-derived growth factor containing two B subunits, EC: endothelial cell; CMs: Cytomodulins, GAG: Glycosaminoglycan; SDF-1: Stromal cell-derived factor 1, SDF1-ELP: SDF1 elastin-like peptide, CK2: casein kinase II, DSS: Asparagine–Serine–Serine, DPP: Dentin phosphoprotein, 3NSS: Asparagine–Serine–Serine, DMP-1: dentin matrix protein 1, HAP: hydroxyapatite, LRAP: leucine-rich amelogenin peptide, PSC: pre-cartilaginous stem cell. The upward arrow indicates an increase in activity. The downward arrow indicates decreased activity.

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
