# Peer review of "Peptide-Based Biomaterials for Bone and Cartilage Regeneration"

_biomedicines, 2024, doi:10.3390/biomedicines12020313_

Round 1
Reviewer 1 Report
Comments and Suggestions for Authors
Dear Editor and Authors,
I would like to express my gratitude for the opportunity to review the manuscript titled "Peptide-based Biomaterials for Osteochondral Tissue Regeneration." I appreciate the chance to contribute to the evaluation of this significant work in the field of regenerative medicine.
After carefully examining the content, I have a few insights that I hope will constructively contribute to the enhancement of the manuscript. The research presented aligns with current trends in osteochondral tissue regeneration; however, I have some suggestions:
- Please state a clear objective in the introduction and ensure it is addressed in the conclusion, which is currently lacking.
- The manuscript is excessively lengthy, totaling around 17,000 words, which makes it almost unreadable. An average review paper typically comprises one-third of this word count. Therefore, please consider condensing repetitive information to enhance readability, aiming to reduce the overall content by half.
- Introduce contradictory ideas to facilitate a discussion of the best options for further depth and insight.
- Improve the quality of the figures for enhanced clarity.
- For Figure 1, enhance the magnification of chondrocytes and lacunae to clarify each. Additionally, include a legend for abbreviations.
Finally, I have a question: why didn’t you performed a systematic review?
Comments on the Quality of English LanguageConsider breaking down longer sentences into shorter ones for improved clarity and readability.
Eliminate redundant phrases and unnecessary repetition to improve the overall flow.
Ensure consistent use of terminology, especially with acronyms like "OCD" or "OC defects."
Author Response
We thank the reviewer for his positive and constructive comments on our manuscript. All points raised by the reviewers in the revised version of this Progress report have been addressed. All changes to our manuscript have been highlighted in the main manuscript file.
1. Please state a clear objective in the introduction and ensure it is addressed in the conclusion, which is currently lacking.
Answer: Thank you for your valuable suggestion. We have revised the introduction by providing a clear objective and clarifying it in the summary and outlook section.
2. The manuscript is excessively lengthy, totaling around 17,000 words, which makes it almost unreadable. An average review paper typically comprises one-third of this word count. Therefore, please consider condensing repetitive information to enhance readability, aiming to reduce the overall content by half.
Answer: We have reduced the total word count of the body of the manuscript to 7685 words (including table and figures), without references. The repetitive information was deleted as much as possible.
3. Introduce contradictory ideas to facilitate a discussion of the best options for further depth and insight.
Answer: Based on the reviewer’s suggestion, the authors attempted their best to rewrite the discussion part in summary and outlook.
4. Improve the quality of the figures for enhanced clarity.
Answer: The quality of the figures was improved to our level best.
5. For Figure 1, enhance the magnification of chondrocytes and lacunae to clarify each. Additionally, include a legend for abbreviations.
Answer: The above issues are carefully addressed in the image as well as legend.
Additional comments:
1) Why didn’t you perform a systematic review?
Answer: Of course, this would have been a better idea. However, due to the time and other constrains, we wish to publish a much better and systematic review on the same in near future.
2) Consider breaking down longer sentences into shorter ones for improved clarity and readability.
Answer: We have thoroughly revised the manuscript for correcting such errors.
3) Eliminate redundant phrases and unnecessary repetition to improve the overall flow.
Answer: As per your recommendation, we have made every effort to eliminate as many repetitions and redundant phrases as possible.
4) Ensure consistent use of terminology, especially with acronyms like "OCD" or "OC defects."
Answer: The same is thoroughly scrutinized, and ambiguities between OC and OCD are fully eliminated in the entire manuscript.
Reviewer 2 Report
Comments and Suggestions for Authors
This review article summarized progress in using peptides derived from different functional molecules of natural full protein for bone and cartilage tissue regeneration. The review was well written, literatures are updated and summarized table are very useful. I recommend accepting with minor revision. Following are my comments:
1. Outline can be removed.
2. Title may be better: “Peptide-based biomaterials for bone and cartilage regeneration”. To fit the contents of the review. Osteochondral is easily misunderstood as osteochondral defect regeneration which is mainly cartilage despite subchondral bone healing is critical for cartilage repair.
3. Line 321, is 100mg/ml concentration correct? That is very high concentration for a functional BMP2 peptide. BMP2 itself can promote chondrogenic differentiation at 50ng/ml concentration according to my work. Please check original literature for accuracy.
4. Adapted figures are blurry.
5. Line 383 “ Due to” was repeated. Please remove one.
6. Table 1, “Source” may be changed to “source molecule”.
7. Line 570, “Bone healing” should be “bone healing in”.
Author Response
We thank the reviewer for his positive and constructive comments on our manuscript. We have addressed all points raised by the reviewer in the revised version of this Progress report.
1. Outline can be removed.
Answer: The outline (content) is removed from the manuscript.
2. Title may be better: “Peptide-based biomaterials for bone and cartilage regeneration”. To fit the contents of the review. Osteochondral is easily misunderstood as osteochondral defect regeneration which is mainly cartilage despite subchondral bone healing is critical for cartilage repair.
Answer: The suggested title is incorporated in the manuscript.
3. Line 321, is 100mg/ml concentration, correct? That is very high concentration for a functional BMP2 peptide. BMP2 itself can promote chondrogenic differentiation at 50ng/ml concentration according to my work. Please check original literature for accuracy.
Answer: Authors deeply regret for the typographic error. It will be 100 µg/mL, which is rectified in the text.
4. Adapted figures are blurry.
Answer: The quality of the figures was improved to our level best.
5. Line 383 “Due to” was repeated. Please remove one.
Answer: The unintended typo is removed.
6. Table 1, “Source” may be changed to “source molecule”.
Answer: The same is changed in Table 1.
7. Line 570, “Bone healing” should be “bone healing in”.
Answer: The same is corrected as per the reviewer’s suggestion.
Reviewer 3 Report
Comments and Suggestions for Authors
In the manuscript titled Peptide-based biomaterials for osteochondral tissue regeneration, Kapat et. al proposed a comprehensive review of various types of peptides dedicated to treating osteochondral defects. The study is well-designed with a logical structure and numerous intriguing examples provided. I would ask the Authors to consider minor modifications to the manuscript before publication:
1. In the Figure 1, please explain the abbreviation of HA for the readers. Moreover, it is type III collagen most abundant in articular cartilage – please comment on it. Finally, explain the type of cells included in that Figure.
2. The quality of the Figures, their resolution, and the font inside should be improved.
3. In Table 1, please add the type of study carried out (in vitro/in vivo) with animal model or cell line culture if applicable.
4. Did the Authors find more information about peptides combined with biomaterials that promote tissue mineralization in terms of hydroxyapatite formation? A separate chapter related to that issue would also be valuable.
5. In my opinion, it would be valuable to add a chapter or table in which the proposed route of administration, dosage form (such as hydrogel, scaffold, sponge), and materials used during manufacture will be included. A short comment on the clinical insertion procedure should be also added.
Regards,
The English language is understandable.
Author Response
The authors thank the reviewer for his positive and constructive comments. We have addressed all points raised by the reviewer in the revised version of this Progress report.
1. In the Figure 1, please explain the abbreviation of HA for the readers. Moreover, it is type III collagen most abundant in articular cartilage – please comment on it. Finally, explain the type of cells included in that Figure.
Answer: The abbreviation of HA is explained in the legend of Fig. 1. Authors deeply regret for the typo for Col III, it would be Col II which is now corrected in the present image. The cells are properly marked in the image as well as in the legend.
2. The quality of the Figures, their resolution, and the font inside should be improved.
Answer: The quality of the figures and font size were improved to our level best.
3. In Table 1, please add the type of study carried out (in vitro/in vivo) with animal model or cell line culture if applicable.
Answer: The existing table 1 was updated with the required information, without significantly increasing the word count and length of the manuscript.
4. Did the Authors find more information about peptides combined with biomaterials that promote tissue mineralization in terms of hydroxyapatite formation? A separate chapter related to that issue would also be valuable.
Answer: The authors are highly thankful to the reviewer for his valuable suggestion. A separate paragraph on biomineralizing peptides has been incorporated in 3.2 Biomineralizing peptides section.
5. In my opinion, it would be valuable to add a chapter or table in which the proposed route of administration, dosage form (such as hydrogel, scaffold, sponge), and materials used during manufacture will be included. A short comment on the clinical insertion procedure should be also added.
Answer: As per the suggestion, the information is updated in Table 1, wherever applicable.
Round 2
Reviewer 1 Report
Comments and Suggestions for Authors
The authors responded appropriately to my suggestions, resulting in a substantial improvement in the manuscript. In my opinion, it is now ready for publication.